# Association between surgical volume and failure of primary total hip replacement in England and Wales: findings from a prospective national joint replacement register

Adrian Sayers [iD],[1,2] Fiona Steele,[3] Michael R Whitehouse [iD],[1,4] Andrew Price [iD],[5,6] Yoav Ben-Shlomo,[2] Ashley W Blom[1,4]

For numbered affiliations see end of article.

**Correspondence to**
Adrian Sayers;
adrian.sayers@bristol.ac.uk

## ABSTRACT

**Objective** To investigate the association of volume of total hip arthroplasty (THA) between consultants and within the same consultant in the previous year and the hazard of revision using multilevel survival models.

**Design** Prospective cohort study using data from a national joint replacement register.

**Setting** Elective THA across all private and public centres in England and Wales between April 2003 and February 2017.

**Participants** Patients aged 50 years or more undergoing THA for osteoarthritis.

**Intervention** The volume of THA conducted in the preceding 365 days to the index procedure.

**Main outcome and measure** Revision surgery (excision, addition or replacement) of a primary THA.

**Results** Of the 579 858 patients undergoing primary THA (mean baseline age 69.8 years (SD 10.2)), 61.1% were women. Multilevel survival found differing results for between and within-consultant effects. There was a strong volume–revision association between consultants, with a near-linear 43.3% (95% CI 29.1% to 57.4%) reduction of the risk of revision comparing consultants with volumes between 1 and 200 procedures annually. Changes in individual surgeons (within-consultant) case volume showed no evidence of an association with revision.

**Conclusion** Separation of between-consultant and within-consultant effects of surgical volume reveals how volume contributes to the risk of revision after THA. The lack of association within-consultants suggests that individual changes to consultant volume alone will have little effect on outcomes following THA.

These novel findings provide strong evidence supporting the practice of specialisation of hip arthroplasty. It does not support the practice of low-volume consultants increasing their personal volume as it is unlikely their results would improve if this is the only change. Limiting the exposure of patients to consultants with low volumes of THA and greater utilisation of centres with higher volume surgeons with better outcomes may be beneficial to patients.

**Strengths and limitations of this study**

- ► This is the largest study in the world to explore the association between surgical volume and outcomes in total hip arthroplasty.
- ► We uniquely calculate a time-varying exposure of surgical volume.
- ► We differentiate between-consultant and within-consultant effects using a multilevel Weibull survival model.
- ► The effect of volume is modelled continuously using restricted cubic splines.
- ► We are unable to affirm causality due to the observational nature of the data.

## INTRODUCTION

Centralisation and specialisation in medical care are advocated to optimise a theorised volume–outcome relationship. In arthroplasty, the volume–outcome relationship has been investigated with respect to outcomes including surgical revision,[1–6] mortality,[1 6–9] patient-reported outcome measures (PROMs)[10] and complications[6 11–14] where volume is measured either by surgeon[1–6] or hospital annual volume.[15] Given the technical requirements of arthroplasty, a strong argument for specialisation based on surgical volume and the risk of revision arthroplasty would exist if volume was causally related to outcome. The evidence to support this assertion is surprisingly sparse and has methodological limitations.[1–6] The principal limitation is the failure to distinguish between-consultant and within-consultant effects.

Differentiating between-consultant and within-consultant effects is crucial to interpreting the data. A between-consultant effect is essentially a cross-sectional analysis that compares the performance of

**BMJ**

one consultant against another and is highly likely to be confounded by centre-level effects.[16] A within-consultant effect is based on individual time series data and compares the changes of volume across time within the same consultant. Correspondingly, within effects can be interpreted more strongly, as the effect of changing a consultant's personal volume, assuming centre-level factors remain relatively constant over the short-term analysis period.

The concept of between-effects and within-effects is well known in epidemiology, and the ecological fallacy is one example. For example, in a standard (single-level) regression analysis, we may observe a positive association between red meat consumption and life expectancy across several countries with differing levels of development. A between-decomposition and within-decomposition would reveal a positive between-country correlation, which is explained by the level of development, and a negative within-country association, that is, individuals within a country who eat more red meat have a lower life expectancy. The decomposition of the volume effect into a between-effect and within-effect is similar, that is, the between-consultant effect is explained by factors, other than volume, which are intrinsic to those consultants and the hospital where they are based ('centre effects'), whereas the within-consultant effect represents the consequence of consultants individually changing their personal volume assuming that other factors remain constant.

In order to facilitate a between-effect and within-effect analysis, consultant volume needs to be assessed continuously across time. Allowing volume to vary over time is computationally intensive but responds to variation in demand and capacity to deliver arthroplasty, specialisation or diversification of professional practice, and is in contrast to previous approaches.[1 4] Additionally, as consultant volume changes, dichotomising the data using arbitrary thresholds, for example, 12 operations per year,[3] is difficult to justify as a consultant's volume will vary over the time they are observed. Furthermore, the interpretation of the results is less likely to be distorted by the arbitrary placement of the threshold.[17 18]

The aim of this research is to investigate the between-consultant and within-consultant (surgeon) effect of the volume of primary total hip arthroplasty (THA) and the risk of subsequent revision.

## METHODS
Using data from the National Joint Registry (NJR) of England, Wales, Northern Ireland and the Isle of Man, we investigated the association between consultant surgical volume in the year (365 days) prior to the index operation of interest, and the risk of revision in patients undergoing elective THA between 1 April 2003 and 22 February 2017.

## Data source
The NJR commenced data collection in April 2003; at inception, it was mandatory for all THA conducted in the private sector to be entered into the NJR, and from 2011, all THA procedures in the public and private sectors were required to be entered into the NJR. A recent national audit of data entered into the NJR between 2014 and 2015 estimated data capture of 95% for primary THA and 91% for revision THA.

## Patient and public involvement
Patient representatives sit on the committee structure of the NJR. The research priorities of the NJR are identified by this committee structure and approved by the patient representatives. Patients were not involved in the setting of the research question or the outcome measures nor were they involved in designing or implementing this work or interpretation of the results. We are unable to disseminate the results of this study directly to study participants due to the anonymous nature of the data. We plan to disseminate our findings to the NJR, via their communications team, to relevant individuals who determine the provision of joint replacement and to the general population through the local and national press.

## Inclusion/exclusion criteria
All consenting patients undergoing THA were eligible for inclusion in the analysis. Patients were included if their patient history was unique and consistent, that is, contained no duplicates, ipsilateral revision prior to the primary or currently held in query by the submitting unit. Due to the requirement of reliable date information, patients who were indicated to have died prior to undergoing a procedure were more than 110 years of age, had undergone a procedure prior to their date of birth or received a procedure prior to 2003 were excluded. Only primary THA, where the sole indication for operation was osteoarthritis (OA) with unique prosthesis combinations, was included. All metal-on-metal bearing combinations were excluded from the analysis due to the known exceptionally high failure rate in this group.[19 20] Consultants with less than 365 days of data were excluded as patients who were less than 50 years of age at the date of the index THA, because these cases are highly likely to be due to secondary OA. See figures 1 and 2 for a detailed breakdown of inclusion criteria.

## Primary outcome
The primary outcome of interest is all-cause revision after a primary THA. Revision arthroplasty was identified by the inclusion of a revision-specific data upload after a primary ipsilateral THA. We note that it is not always the primary surgeon that performs the revision.

## Censoring
Patients were censored following death. Death status was established by linking patients to the National Health Service (NHS) Personal Demographic Tracing Service.

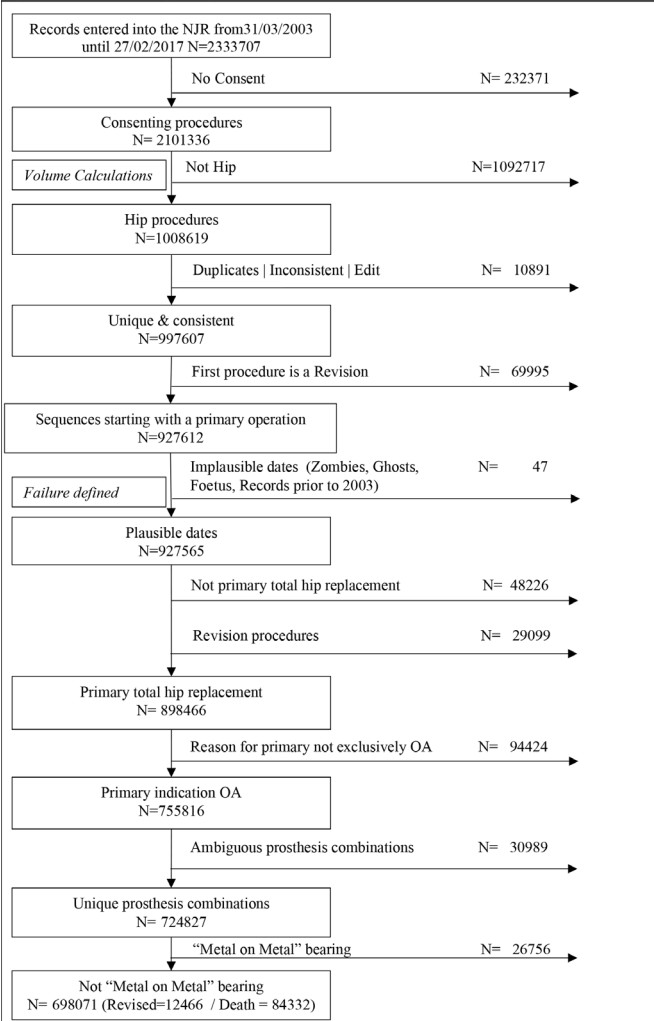

**Figure 1** Flowchart of the participant inclusion and exclusion criteria. NJR, National Joint Registry; OA, osteoarthritis.

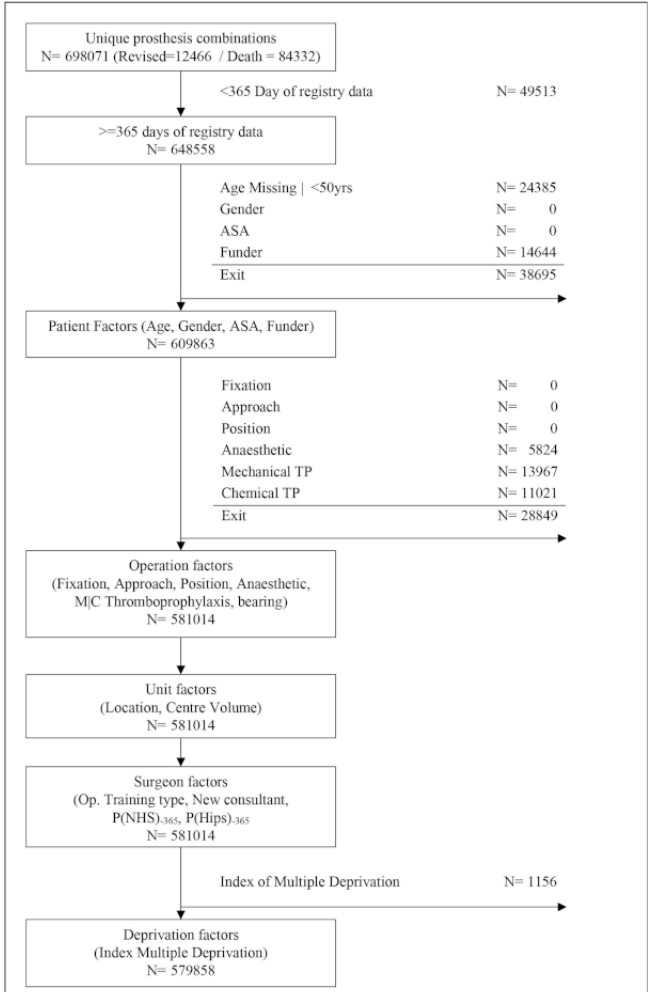

**Figure 2** Flowchart of the participant missing data. ASA, American Society of Anesthesiologists; MC, Mechanical or Chemical; NHS, National Health Service; TP, Thromboprophylxaxis.

## Primary exposure

The primary exposure of interest in this study was the consultant surgical volume of any THA recorded in the NJR in the preceding 365 days prior to the index procedure in consenting patients, and prior to the application of inclusion and exclusion criteria, see figure 1. We choose a 365-day period as this represents one calendar year, and this effectively integrates out seasonal variation from the volume definition,

## Confounding factors

Confounding factors were thematically organised into five groups:

1. Patient factors: age, sex, American Society of Anesthesiologists (ASA) grade and operation funder.
2. Operation factors: THA fixation, approach, patient position during arthroplasty, anaesthetic type, thromboprophylaxis regime, bearing and year of primary THA.
3. Centre factors included the setting of the treatment episode (ie, private or NHS hospital) and surgical centre volume.

4. Consultant-based factors included the training status of the primary surgeon performing the operation, whether the responsible consultant was listed in the NJR after 2008 (ie, a newly qualified surgeon), the proportion of THA undertaken in the NHS in the preceding 365 days by the consultant (ie, a public or private hospital surgeon), the proportion of THA procedures undertaken in the previous year compared with all joints recorded by the NJR (ie, a specialist hip surgeon or a general arthroplasty surgeon).
5. Deprivation factors were based on the English and Welsh indices of multiple deprivations, an area-based index of patient socioeconomic status. See online supplementary table 1.

## Statistical analyses

Means, SDs and interquartile points were used to describe continuous variables. Frequencies and percentages were used to describe categorical variables. The association between confounding factors and consultant volume was explored by comparing summary statistics between levels of each factor.

Graphical methods including frequency distributions and empirical cumulative distributions were used to describe the relative frequency and centiles of the volume distributions. The empirical cumulative distribution allows centiles of the distribution to be quickly identified.

The associations between consultant volume in the preceding 365 days of the index procedure and all-cause revision were explored using multilevel parametric (Weibull) survival models.

Between-effects and within-effects are decomposed using a process known as group mean centring.[21 22] Group mean centring is the process of creating two new variables from the primary exposure. The first variable is the consultant-specific mean volume and the second variable is the deviation of a consultant's volume for a given procedure from their personal average (ie, consultant's mean-centred volume). The between-effect is estimated as the coefficient of a consultant's mean volume, whereas the within-effect is the coefficient of the consultant's deviation from their average.

Continuous variables were modelled used orthogonalised restricted cubic splines (RCS), this also included the volume effect. Each RCS is centred at meaningful value, these values are listed in online supplementary table 1, mean between-consultant volume is centred at 32 procedures annually, whereas within-consultant volume is naturally centred at zero. We iteratively varied the number of knot points and placement strategies, in unadjusted models to optimise fit. The most parsimonious specification of RCS was selected using Akaike information criterion.[23]

Confounding adjustment was conducted incrementally introducing patient, operation, centre, surgeon, and finally, deprivation confounding variable groups. The effect of confounding adjustment on the primary exposure of interest was explored and presented at each stage of the model building process allowing the effect of adjustment to be clearly illustrated. All modelling was conducted using the *mestreg* package in Stata V.15.1.[24] The specification of the model is described in more detail in the supplementary material (section- Multilevel Weibull model).

### Missing data

Given the large data set and small fraction of incomplete cases among the observed (89% of all eligible records were included in the analysis), any improvement in efficiency from multiple imputations is likely to be negligible, and a complete-case analysis would provide unbiased results.[25 26] Therefore, we have assumed that the reason for the missingness is independent of both the outcome and primary exposure of interest.

### RESULTS

Between 1 April 2003 and 22 February 2017, 1 008 619 primary and revision THA procedures were entered in the NJR. After application of inclusion and exclusion

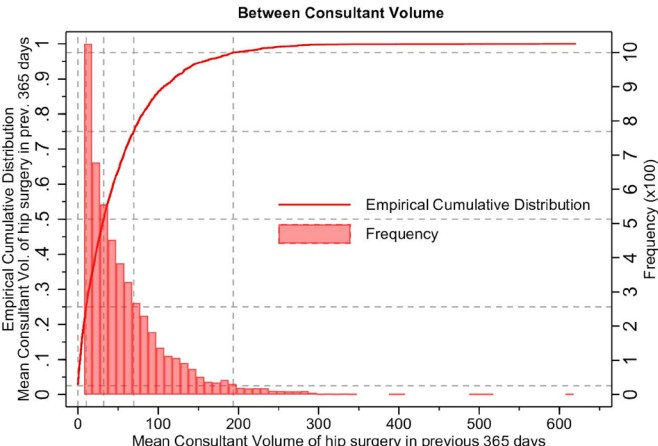

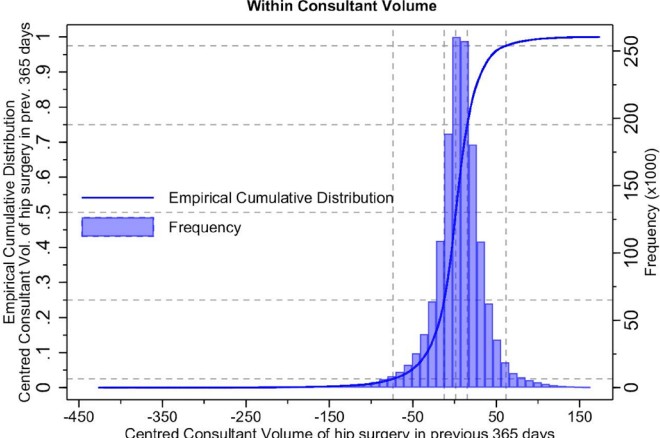

**Figure 3** Empirical cumulative distribution and frequency distribution of (between) mean consultant volume and (within) individual centred volume of hip arthroplasty in the previous 365 days. Grey horizontal hashed lines indicate the 2.5th, 25th, 50th, 75th and 97.5th centiles of the distribution, vertical hashed lines indicate mean and centred consultant volume at 2.5th, 25th, 50th, 75th and 97.5th centiles, respectively.

criteria, 579 858 primary THA replacement procedures were available for analysis, (figure 1), including 9238 revised procedures, with 2.5 million years of observation time, with a maximum length of follow-up of 13.14 years. Patients were predominately women (61.1%), ASA II (70.4%), treated in public (NHS) hospitals (84.1%) and had a mean age 69.8 years (SD 10.2). See online supplementary tables 2 and 3 for full descriptive statistics of the population of patients used in the analysis.

Mean between-consultant volume in the previous year was positively skewed, with only a minority (>97.5% of the empirical cumulative distribution) of consultants performing in-excess of 200 procedures a year (figure 3). The median number of procedures conducted across all consultants, for the whole analysis period, in the year prior to an index procedure was 95 (IQR 52–158) with substantial variability between consultants, (figure 4). Mean within-consultant volume, for the most part, was symmetrically distributed. There was substantial variability of volume within consultants (median within-consultant

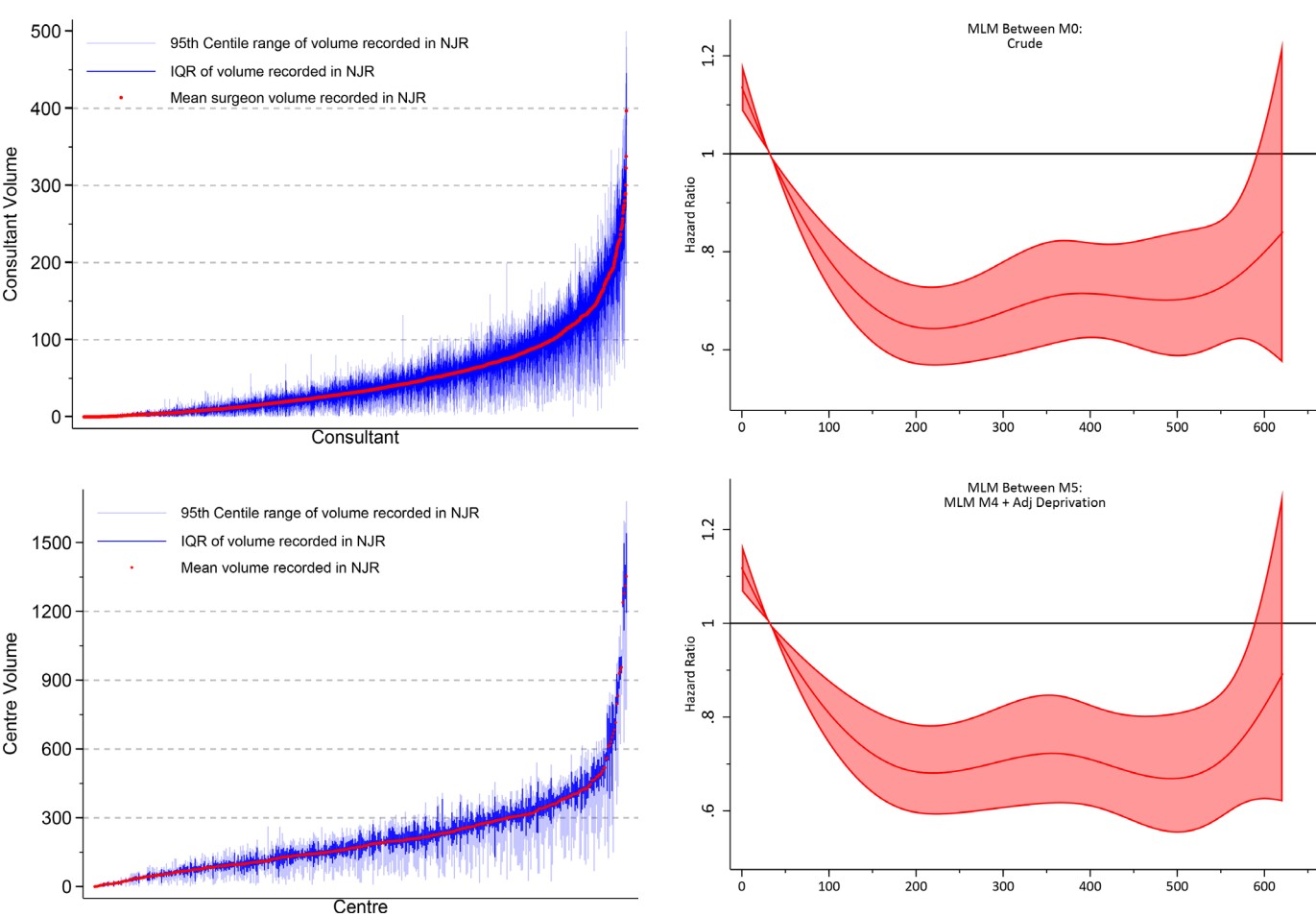

**Figure 4** Mean, IQR and 95th centile range of consultant and centre volume of hip arthroplasty in the previous 365 days recorded in the NJR by individual consultant and individual unit, respectively. NJR, National Joint Registry.

**Figure 5** Between-consultant marginal association of hip surgical volume in the preceding 365 days and hazard of revision arthroplasty unadjusted (M1) and adjusted (M5) for confounding factors in a multilevel model (MLM). Patient factors include sex, AmericanSociety of Anesthesiologists grade and funder. Operation confounding factors include fixation, approach, position, anaesthetic, mechanical and chemical thromboprophylaxis, bearing and year of operation. Centre confounding factors include hospital location and centre volume in the preceding 365 days. Surgeon confounding factors included lead operating surgeon, listing of a surgeon within National Joint Registry prior to 2008, the proportion of National Health Service cases in the preceding year and proportion hip arthroplasty procedures undertaken in the previous year.

SD is 23 (IQR 14.5–36.5)), (figure 3) and consultants with larger mean volumes tended to have greater variability (figure 4). We also note that the IQR of centre volume is much less variable than consultant volume but is more negatively skewed at centre level (figure 4).

Summary statistics of consultant volume by confounding factors at the index procedure are listed in online supplementary table 4. Higher volume consultants were observed to treat patients who were younger, with lower ASA grade, who privately funded their THA and received a uncemented or hybrid prostheses more frequently. They were also more likely to use a posterior approach, place patients in the lateral position, work in the private sector, predominately perform hip arthroplasty and treat patients from more deprived areas (see online supplementary table 4).

The effect of confounding adjustment on the volume association was explored using complete cases in estimation of the multilevel Weibull model. The marginal between-consultant and within-consultant effects of volume in the preceding year on hazard of revision and 95% CI are presented in figure 5, online supplementary figures 1 and 2, respectively.

A near-linear 48.8% (95% CI 36.3 to 61.4) reduction in HR was observed in crude models for volumes between 1 and 200 procedures annually, that is, from 1.13 (95% CI 1.08 to 1.18) to 0.64 (95% CI 0.56 to 0.72), with a flattening of risk beyond this, noting that only ~2.5% of consultant recorded more than 200 procedures annually. Despite extensive confounding adjustment, the marginal association of the between-consultant volume effect remained remarkably consistent across the range of volumes observed. In our fully adjusted (deprivation) model, a near-linear 43.3% (95% CI 29.1 to 57.4) reduction in HR was observed. The marginal association of the within-consultant volume effect and revision was similarly

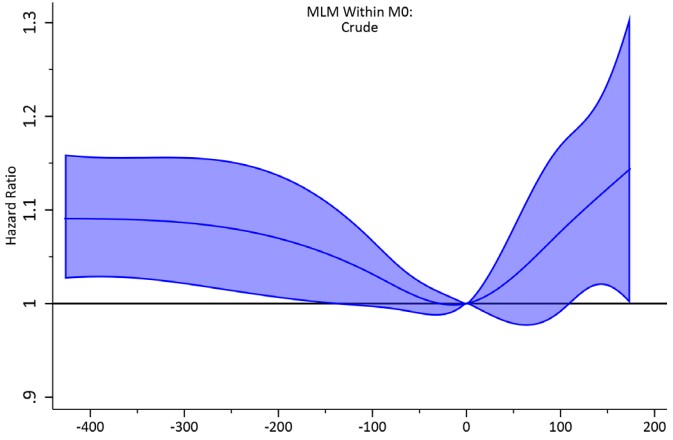

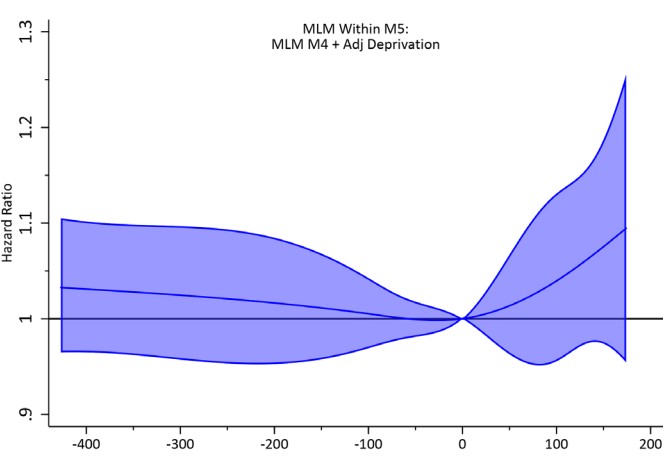

Consultant volume of hip surgery in previous 365 days

**Figure 6** Within-consultant marginal association of hip surgical volume in the preceding 365 days and hazard of revision arthroplasty unadjusted (M1) and adjusted (M5) for confounding factors in a multilevel model (MLM). Patient factors include sex, AmericanSociety of Anesthesiologists grade and funder. Operation confounding factors include fixation, approach, position, anaesthetic, mechanical and chemical thromboprophylaxis, bearing and year of operation. Centre confounding factors include hospital location and centre volume in the preceding 365 days. Surgeon confounding factors included lead operating surgeon, listing of a surgeon within National Joint Registry prior to 2008, the proportion of National Health Service cases in the preceding year and proportion hip arthroplasty procedures undertaken in the previous year.

consistent and showed no evidence (figure 6) of an association with revision if consultants increased or decreased their volume by ±50 procedures per year (HR ~1% and 95% CI includes zero).

## DISCUSSION

We provide novel insights into the volume–outcome relationship of 579 858 elective THA patients using a between-decomposition and within-decomposition to analyse the association of consultant volumes on revision. Between different consultants, the volume of arthroplasty in the previous year is associated with a near-linear 49%

and 43% reduction in HR, between 1 and 200 procedures with revision THA in crude and fully adjusted models, respectively. Within the same consultant, we demonstrate that there is no evidence of an association between volume of THA in the previous year and risk of revision.

Uniquely, we use a time-varying volume specification that facilitates the decomposition of between-consultant and within-consultant effects. We suggest the within-consultant effect is much closer to the causal interpretation desired by many policymakers, and failure of research to recognise the difference among between-effects and within-effects may lead to erroneous policy decisions and unintended consequences.

We demonstrate that optimal between-consultant results are reached when the consultant volumes in the previous year are approximately 200 procedures. We suggest that these factors are not causally related to volume, but rather due to unmeasured surgeon, patient and/or centre factors. There is no evidence to suggest that consultants should change their personal volume in the hope of improving their outcomes or that there is an arbitrary threshold where the outcome of results becomes good. It is important to add that these volume changes are for experienced consultants who have already passed early improvements that one might observe at the early trainee level due to practice. Furthermore, their low volume for hip replacement may reflect high volume experience for other procedures, thereby ensuring manual dexterity although for a different operation.

While the results appear contradictory compared with previous research, that is, no threshold volume effect, the differences may be explained by the method of analysis, that is, single-level models versus multilevel model, and interpretation of results, that is, separation of between-consultant and within-consultant volume effects. Previous analyses are single-level analyses and assume that all procedures are independent of one another, so that a low-volume consultant would achieve the results of a high-volume consultant if they could instantly increase their personal volume. The interpretation of a single-level model is similar to that of the between-consultant interpretation. While this may be an attractive interpretation for policymakers, it fails to recognise the complexity of the data and processes observed, and that there are many factors, intrinsic to each consultant and the centre or centres in which they work that predispose them to be either low-volume or high-volume surgeons, for example, fellowship trained, threshold for revision or unit organisation.

This study has a number of strengths and limitations. Strengths include: (1) our unique decomposition of between-consultant and within-consultant which we believe provides a more actionable interpretation for policymakers. (2) Time-varying consultant volume, which is independent of the index procedure, allows for stronger inferences. (3) OA was the only indication for arthroplasty, which we believe represents a 'best-case scenario', and the volume effect will only be attenuated by the

inclusion of other diagnoses. (4) We demonstrate the use of RCS to model volume effects, which ensures flexibility and a smooth continuous function, emphasising the lack of any threshold in the volume effect. (5) The study is significantly (10 times) larger than any other published study on this topic,[1–6] with a maximal follow-up period of more than 13 years. (6) We have conducted extensive case-mix adjustment and illustrate that both between-effects and within-effects are insensitive to our measured confounding factors.

Despite the many strengths, this study has a number of limitations. (1) The analysis and decomposition of between-consultant and within-consultant effects is more complex than traditional analyses assume and requires careful interpretation. (2) Despite the independent nature of the volume of arthroplasty calculation prior to the index procedure, we still see anticipated associations, that is, younger patients, patients with lower ASA scores, patients receiving uncemented implants and patients operated on in the private sector all tend to be treated by higher volume consultants. Consultants specialising in THA and working principally within the private sector accrue higher volumes. These features suggest a strong propensity in practice to treat similar patients and raise the possibility of lagged correlation in unmeasured confounders.[27] (3) The use of a single indication for arthroplasty, namely, OA, may limit the generalisability of results particularly in regards to Black or Asian ethnicities where OA is not as dominant an indication for THA.[28] (4) The use of RCS requires extensive sensitivity analyses to ensure knot points are placed optimally, and that results are not sensitive to knot placement. (5) Calculation of the time-varying volume specification of THA in the previous year is computationally intensive and requires significant parallelisation before analyses can be started. (6) Our covariates are unlikely to capture important centre differences in staffing, organisation and policy in the management of THAs. (7) Volume does not fully encapsulate experience or expertise that may have been acquired prior to the inception of the register, for example, during fellowships or from working in other healthcare systems or the potential persistent effect of learnt experience. (8) Revision surgery is only one possible endpoint and other endpoints including PROMs or other adverse events may also be important to patients. (9) We only considered a multilevel Weibull model to model the baseline hazard; other more flexible functional forms may accommodate the baseline hazard more appropriately, however, model convergence is always a challenge.

We suggest that the within-consultant effect from the multilevel regression is much closer to the causal interpretation required by consultants, patients and policymakers, that is, what is the effect of changes in personal volume on the hazard of revision THA? This is not to say the between-effect is not of interest to policymakers, but to say that the between-effect suggests that there are intrinsic differences between high-volume and low-volume consultants, that is, expertise, where higher volume consultants tend

to have better outcomes, but these differences cannot be attributed to volume per se. We suggest that our analyses illustrate 'State versus Trait' behaviour, where between-consultant association illustrates the 'traits' of surgeons and within-consultant associations illustrate their 'state'. This is to say traits of experienced high-volume surgeons with good outcomes are unaffected by changes to their personal volume. Conversely, experienced low-volume arthroplasty surgeons who transiently increase their personal volume do not improve their outcomes.

## CONCLUSION

In summary, using data from the largest arthroplasty register in the world,[29] we have demonstrated that there is no within-consultant association between surgical volume in the previous year and the risk of revision in patients undergoing primary THA for OA, whereas there is strong evidence to suggest higher volume consultants tend to have better outcomes for reasons that are unlikely to be due to the volume of arthroplasty in the previous year per se.

The results from this study have profound implications for quality improvement within healthcare. Encouraging consultants to undertake a minimum number of procedures under the guise of raising standards could be counterproductive and may only serve to expose patients to increased risk of revision by low-volume or previously low-volume consultants. Centralisation and specialisation of THA in consultants who, for reasons, not including volume, can undertake a greater number of procedures are likely to benefit patients and reduce the revision burden overall. Encouraging or training low-volume consultants to use prosthesis combinations with better outcomes may be more effective methods of improving outcomes for patients.

Importantly, the combined use of an independent time-varying consultant volume with multilevel (between and within) regression modelling allows results to be interpreted more clearly than a cross-sectional analysis of conventional observational data. Our work highlights the importance of appropriate methodology, and while this may fall short of being definitively causal, results from a randomised experiment are unlikely to be feasible in this population. We believe this represents the best available evidence to guide policy formulation.

**Author affiliations**
[1]Musculoskeletal Research Unit, Bristol Medical School, Southmead Hospital, University of Bristol, Bristol, UK
[2]Population Health Sciences, Bristol Medical School, University of Bristol, Bristol, UK
[3]Department of Statistics, London School of Economics and Political Science, London, UK
[4]National Institute for Health Research Bristol Biomedical Research Centre, University Hospitals Bristol NHS Foundation Trust and University of Bristol, Bristol, UK
[5]Nuffield Department of Orthopaedics, Rheumatology and Musculoskeletal Sciences, University of Oxford, Oxford, UK
[6]NIHR Biomedical Research Unit, University of Oxford, Oxford, UK

**Acknowledgements** We thank the patients and staff of all the hospitals who have contributed data to the National Joint Registry. We are grateful to the Healthcare Quality Improvement Partnership, the National Joint Registry Steering Committee and staff at the National Joint Registry for facilitating this work. The views expressed represent those of the authors and do not necessarily reflect those of the National Joint Registry Steering Committee or Healthcare Quality Improvement Partnership, who do not vouch for how the information is presented.

**Contributors** AS, FS, YB-S, AP, MRW and AWB were responsible for the study design, AS conducted the data analysis. AS, FS, YB-S, AP, MRW and AWB were responsible for interpreting the data. AS, FS, YB-S, AP, MRW and AWB prepared and edited and approved the final manuscript.

**Funding** AS is funded by an MRC Strategic Skills Fellowship MR/L01226X/1. This study was supported by the NIHR Biomedical Research Centre at University Hospitals Bristol NHS Foundation Trust and the University of Bristol. The views expressed in this publication are those of the author(s) and not necessarily those of the NHS, the National Institute for Health Research or the Department of Health and Social Care. The Medical Research Council or National Institute of Health Research had no role in the design and conduct of the study; collection, management, analysis and interpretation of the data; preparation, review or approval of the manuscript; and decision to submit the manuscript for publication.

**Competing interests** None declared.

**Patient consent for publication** Not required.

**Ethics approval** Pseudo anonymised analysis of NJR data is considered as secondary use of clinical registry data, under HRA guidance, this does not require formal ethical approval. The full NJR privacy notice can be found online (http://www.njrcentre.org.uk/njrcentre/About-the-NJR/Privacy-Notice-GDPR).

**Provenance and peer review** Not commissioned; externally peer reviewed.

**Data availability statement** Data may be obtained from a third party and are not publicly available. Access to the data can be made via research requests to the National Joint Registry of England, Wales, Northern Ireland and the Isle of Man. Full details can be found at http://www.njrcentre.org.uk/njrcentre/Research/Research-requests.

**ORCID iDs**
Adrian Sayers http://orcid.org/0000-0001-7452-5043
Michael R Whitehouse http://orcid.org/0000-0003-2436-9024
Andrew Price http://orcid.org/0000-0002-4258-5866

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
