## [Reviewer comments · BMJ Open]

ARTICLE DETAILS

TITLE (PROVISIONAL)	The association between surgical volume and failure of primary total hip replacement in England and Wales: Findings from a prospective national joint replacement register
AUTHORS	Sayers, Adrian; Steele, Fiona; Whitehouse, Michael; Price, Andrew; Ben-Shlomo, Yoav; Blom, AW

VERSION 1 – REVIEW

REVIEWER	Richard Jenkinson Sunnybrook HSC Toronto Canada
REVIEW RETURNED	14-Aug-2019

GENERAL COMMENTS	This is a big data project using data from the National Joint Registry. The numbers are impressive and the inclusion/exclusion process and management of confounding variables is appropriate. Interesting approach to the data. Very statistically sophisticated and challenging to understand without high level of statistical knowledge. I am an orthopaedic surgeon with training in epidemiology and required a statistical review to confirm the validity of the statistical methods. Assuming the statistics are appropriate, I support publication. The figures are challenging to understand. Thought should be given to ways to present the data that would be more accessible to the target audience of orthopaedic surgeons and administrators. Is there a way to present the data more succinctly? Figure 2 is a simple concept presented in a challenging way. Does the empirical distribution line need to be on the same figure. The frequency distribution presented on its own should be sufficient to display to the reader the variation of hip procedure volumes done while simplifying the figure. Do all 6 between surgeon and within surgeon multilevel models need to be presented especially when they are all very similar? I would find the crude model and then the fully adjusted model sufficient to display your findings. The titles of these particular figures use unclear abbreviations and jargon as labels. The presentation of these models could be improved to ensure that a non-statistical expert can understand the point that the authors are making. I think it important to frame the findings more strongly in terms of the support for high volume centers and surgeons. There is a very strong volume outcome effect shown in this data for higher volume surgeons having better results. It is interesting that no threshold emerged from this data which has been seen elsewhere. The centre volume is presented but it is not specifically analyzed in this paper. This figure does not add anything to this particular paper. A similar analysis looking at the volume of hip arthroplasty at a centre as the
---

	variable related to revision rate would be a natural question flowing from this data. Having centre effects discussed in this paper would greatly strengthen the usefulness of the paper. It is interesting that the yearly variation in hip arthroplasty numbers for a surgeon do not affect revision rates but the important questions are what makes a high volume surgeon have a lower revision rate. This paper does not answer that question but provides new and interesting information to inform policy on provision of hip arthroplasty.
--	--

REVIEWER	Andreas Halder Sana Kliniken Sommerfeld Germany
REVIEW RETURNED	06-Oct-2019

GENERAL COMMENTS	pros  - interesting research question - large database from National Joint Registry - elaborate statistical analysis - excellent English language Cons  - Statistical analysis method needs to be thoroughly reviewed by experienced statistician - Statistical method description hard to understand for non-statisticians and dominates the orthopaedic content at some points - As only procedures 365 days prior to index procedure were analysed, “between consultant” effects were significant as practice from surgeon to surgeon is different, but “within consultant effects” were of course not significant as experience and practice of a surgeon does not change within a year even if the volume increases in this time period. P5 L 36: “A within-consultant effect is based on a time series of individual data and compares changes of volume across time within the same consultant.” – but that needs analysis of a longer period of time than one year. Additionally, a decrease in surgical volume over time is likely to have a different effect than an increase as an experienced surgeon could treat a lower number of patients with a low revision rate while an unexperienced surgeon increasing in a short period of time still keeps a higher revision rate. (P13 L 29, P 14 L 37, P 16 L 39, L 57). Therefore, the conclusion, that volume has no effect on surgical quality and in consequence revision rate is not fully justified. - Comparison of between consultant effects of 1 to 200 procedures annually is quite rough and statistical significance is not surprising - P 8 L 36 all cause revision as primary outcome is probably not suitable as soft tissue revision for hematoma is more depending on surgeon’s preferences than surgical quality and is very different from revision for dislocation or infection - P 15 L 38 follow up is not really 13 years overall! Specific comments  - P 1 L 11 “hazard of revision” - P 2 L 21 the findings are not really “novel” - P 5 L 29 “between between” - P 12 L 15 “13.14_ years” - P 12 L 17 ”and treated public (NHS) hospitals (84.1%), had a mean age of 69.8 years” - P 16 L 48 no, the National Joint Registry is not the largest in the world!
--

REVIEWER	Sameer Naranje UAB USA
REVIEW RETURNED	16-Oct-2019

GENERAL COMMENTS	A very well formulated study. However, authors have not compared the results of their study to the existing similar studies in the discussion section and needs to be revised. ", " after before reference citations should be removed.
--

REVIEWER	Kim Madden McMaster University, Canada
REVIEW RETURNED	05-Nov-2019

GENERAL COMMENTS	This is a very interesting an important topic in orthopaedics. The authors have taken great care in conducting this study in a more methodologically stringent way than previous studies and for that I applaud them. It is well-written and reported according to STROBE standards. I do have a few conceptual questions that I am hoping the authors can respond to:  1. For within consultant effects, how do you distinguish whether the volume or the expertise is affecting the outcomes? Although they are related, I think expertise and volume at a particular point in time are two separate concepts. 2. Second line of the discussion: the authors state "...to analyse the effects of consultant volumes on revision...". I think this should be association not effects as the authors themselves acknowledge that they cannot establish a causal relationship. 3. While I agree with the authors that "...these factors are not causally related to volume, but rather due to unmeasured factors such as centre effects.", I don't think it necessarily follows that surgeons should not increase their volume to improve outcomes. What about expertise effects? If surgeons do more cases they will become better at those cases over time. However, if high-volume surgeons decrease their volume (within reason) they will not necessarily lose that level of expertise. I don't think this study has adequately addressed the concept of expertise changing over time. 4. The authors only considered revision in their analyses and not any other outcomes. This is a limitation, particularly because threshold for revision can change over time as surgeons gain expertise and experience. 5. The authors state in the conclusion that "Encouraging consultants to undertake a minimum number of procedures under the guise of raising standards could be counterproductive and may only serve to expose patients to increased risk of revision by low or previously low volume consultants". This might be true if the surgeons have little interest in or aptitude for increasing their volume. However, I think expertise is affected by both surgeon factors and volume, which isn't really addressed here.
---

REVIEWER	Zhiying You University of Colorado Anschutz Medical Campus
REVIEW RETURNED	06-Nov-2019

GENERAL COMMENTS	It is a very interesting study with potential implication of policy making. Addressing the following concerns could make it more interesting to readers.  1. Provide more details or more clear definition of the variables
---

	involved in the analysis wherever appropriate. 2. Page 10, line 10, while the Weibull distribution was specified for analysis, readers may wonder what if other distributions had been assumed. Furthermore, adding some more details of the survival model used would help readers understand better the methods and make it possible that readers can replicate the analysis for given data. 3. Page 10, line 55, while it was mentioned that “Given the large data set and small fraction of incomplete cases among the observed (FICO), any improvement of efficiency from multiple imputation is likely to be negligible, and a complete-cases analysis would provide unbiased results”, readers may want to know how small the fraction was. Providing the number will be better. 4. Page 14, line 39, it was mentioned “rather due to unmeasured factors such as centre effects”. Probably it should also include unmeasured physician and/or patient factors. 5. Pages 25-28, there are needs of markers numbers for the figures, e.g. figure 3, figure 4. Right now readers don’t know which figure was referred to when reading Figure 2 and Figure 3.
--	---

VERSION 1 – AUTHOR RESPONSE

Reviewer: 1 “Richard Jenkinson”

We would like to thank Richard Jenkinson for taking the time to review our work.

This is a big data project using data from the National Joint Registry. The numbers are impressive and the inclusion/exclusion process and management of confounding variables is appropriate.

Interesting approach to the data. Very statistically sophisticated and challenging to understand without high level of statistical knowledge. I am an orthopaedic surgeon with training in epidemiology and required a statistical review to confirm the validity of the statistical methods. Assuming the statistics are appropriate, I support publication. The figures are challenging to understand. Thought should be given to ways to present the data that would be more accessible to the target audience of orthopaedic surgeons and administrators. Is there a way to present the data more succinctly?

Figure 2 is a simple concept presented in a challenging way. Does the empirical distribution line need to be on the same figure. The frequency distribution presented on its own should be sufficient to display to the reader the variation of hip procedure volumes done while simplifying the figure.

We agree with the reviewer that the data in this papers are challenging. We have found it very difficult to accurately and transparently display important data, that can be used to motivate the analysis.

The empirical cumulative distribution allows median, interquartile points or any other centile to be quickly identified / displayed. In essence we present a box and whisker plot which accurately describes the percentiles of the distribution.

Given how many surgeons claim to be high volume it is somewhat surprising that the median number of procedures is less than 1 joint a week, and furthermore less than 2.5% of all the surgeons in England and Wales have carried out more than 200 joint replacements a year.

*The Empirical cumulative distribution helps to quantify and visually represent this.
[Page 10]*

Do all 6 between surgeon and within surgeon multilevel models need to be presented especially when they are all very similar? I would find the crude model and then the fully adjusted model sufficient to display your findings.

Our intention was to illustrate how insensitive to adjustment the effects of interest are. Controlling for confounding can be somewhat contentious, with some individuals arguing that only patient factors should be adjusted for, whilst others argue that operative characteristics should be included. Therefore, we included a broad range of adjustment characteristics to allow readers to see the consequences of adjustment.

However, we have now moved models 2 to model 4 into the supplementary material

We hope this makes things clearer. [Page 10]

The titles of these particular figures use unclear abbreviations and jargon as labels.

The main figure legends are included in the manuscript, and we can see how the sub-headings on their own would be slightly confusing. We have added more information to the main legend to clarify the abbreviations. [Page 20, 21]

The presentation of these models could be improved to ensure that a non-statistical expert can understand the point that the authors are making.

We thank you for your comments. We have found it difficult to convey what are non-linear associations between exposure and outcome without the use of graphical methods. We appreciate the concept of within- and between- effects is challenging. However, we do not know of any other way which can as succinctly display all the crucial information. Any simplification or alternative parametrisation of the data may lead to false conclusions, including the presence of thresholds for example.

I think it important to frame the findings more strongly in terms of the support for high volume centers and surgeons. There is a very strong volume outcome effect shown in this data for higher volume surgeons having better results.

We have now strengthened the conclusions that support the centralisation of care for surgeons who are conducting large volumes as they appear to get better results. However, as we have not specifically investigated the association between centres and volume, we do not believe we can suggest in our conclusions that centralisation in large volume centres is warranted. [Page 17]

It is interesting that no threshold emerged from this data which has been seen elsewhere.

We believe the threshold effect that many papers refer to is artificially created by dichotomising the data, and failing to consider the process in enough detail, and other fundamental methodological problems i.e. distinguishing between- and within- surgeon association. We now highlight this in the discussion. [page 14]

The centre volume is presented but it is not specifically analyzed in this paper. This figure does not add anything to this particular paper.

We used centre volume as an adjustment factor in the analysis, and given the complexity of describing volume across the units, we believe this is the most succinct method in which we can describe this important confounding factor. We also note that centre volume is not normally distributed with long left tails, suggesting variation in the provision of hip replacement in the country. We now comment on this. [Page 12]

A similar analysis looking at the volume of hip arthroplasty at a centre as the variable related to revision rate would be a natural question flowing from this data. Having centre effects discussed in this paper would greatly strengthen the usefulness of the paper.

We agree that this is an interesting and obvious extension to this work. However, given the complexity of the data and the methods of the analyses presented here already, we believe this is likely to overload the paper.

It is interesting that the yearly variation in hip arthroplasty numbers for a surgeon do not affect revision rates but the important questions are what makes a high volume surgeon have a lower revision rate. This paper does not answer that question but provides new and interesting information to inform policy on provision of hip arthroplasty.

We would like to thank the reviewer for taking the time in reading and commenting on our paper.

Reviewer: 2 (Andreas Halder)

We would like to thank Andreas Halder for taking the time to review our work.

Please leave your comments for the authors below pros

- interesting research question
- large database from National Joint Registry
- elaborate statistical analysis
- excellent English language

We thank you for your interest in our work

Cons

- Statistical analysis method needs to be thoroughly reviewed by experienced statistician
- Statistical method description hard to understand for non-statisticians and dominates the orthopaedic content at some points
- As only procedures 365 days prior to index procedure were analysed, “between consultant” effects were significant as practice from surgeon to surgeon is different, but “within consultant effects” were of course not significant as experience and practice of a surgeon does not change within a year even if the volume increases in this time period. P5 L 36: “A within-consultant effect is based on a time series of individual data and compares changes of volume across time within the same consultant.” – but that needs analysis of a longer period of time than one year. Additionally, a decrease in surgical volume over time is likely to have a different effect than an increase as an experienced surgeon could treat a lower number of patients with a low revision rate while an inexperienced surgeon increasing in a short period of time still keeps a higher revision rate. (P13 L 29, P 14 L 37, P 16 L 39, L 57). Therefore, the conclusion, that volume has no effect on surgical quality and in consequence revision rate is not fully justified.

There are a number of extremely important points you raise.

We agree that there is a difference between experience and volume. We note that volume calculations are based on consultant (attending) surgeons who are working independently within England and Wales. i.e. they should all be experienced enough to be working independently.

We believe experience is probably best characterised by a surgeon’s average surgical volume and represented in the between-surgeon analysis. This is why high average, experienced, volume surgeons have better results than low average inexperienced surgeons. As you suggest, it may be somewhat unsurprising that if a high average volume surgeon reduces their volume the surgical results are unaffected. Similarly, if a low volume, inexperienced, surgeon temporarily increases their personal volume it does not alter their results either, i.e. they still remain relatively inexperienced.

This is somewhat akin to state vs. trait behaviour. And whilst it is technically true, that the surgeon’s average volume, their “trait”, can be modified over an extended period of time, we think that the normally distributed “within surgeon” volume displayed in figure 2, suggests this is unlikely. Furthermore, Figure 3 which essentially shows box and whisker plots for each consultant surgeon in the registry illustrates that their IQR are approximately centred on their personal mean.

[Page 16, 17]

Using a lagged volume, which is continuously recalculated after every procedure, is very computationally intensive. But has a number of important benefits. The first is that the volume is reactive to changes, and the magnitude of any observed change is limited. If you were to perform surgery every day, the lagged volume could only change by the number of surgeries performed in that data. We choose 1 year, as we believed this would characterise behaviour without being unduly influenced by holidays, winter pressures or anything else that may happen in the annual cycle. Choosing a lag of less than a year would mean volume would potentially be influenced by circumstances which are known to have cyclic variation. A 365 day lag effectively integrates seasonal variation. We do not believe that choosing a cycle longer than 1 year will be fruitful due to the lack of responsiveness.

We now comment on this in the methods.[Page 9]

- Comparison of between consultant effects of 1 to 200 procedures annually is quite rough and statistical significance is not surprising

Our reason for choosing 1 to 200 procedures is because the association between these two volumes is approximately linear, and shows the extreme between high and low volume surgeons.

We believe the linearity of the association is surprising, given the previous hypothesized thresholds.

- P 8 L 36 all cause revision as primary outcome is probably not suitable as soft tissue revision for hematoma is more depending on surgeon's preferences than surgical quality and is very different from revision for dislocation or infection

We recognise that other end points may be of interest to patients and surgeons. We now comment on this. We are limited by the data contained in the registry. The NJR defines revision as any procedure which adds, removes or modifies an implant and therefore soft tissue revision procedures are not captured. [Page 16]

We hope this paper provides a framework to investigate volume effects for other outcomes.

- P 15 L 38 follow up is not really 13 years overall!

We have clarified this as a maximum followup. [Page 15]

Specific comments

- P 1 L 11 "hazard of revision"

We have now amended this typo [Page 1]

- P 2 L 21 the findings are not really "novel"

We respectfully disagree. We believe we are first to use a within- and between- decomposition to explore the effect of volume on revision.

- P 5 L 29 "between between"

We have now deleted the extra "between"

- P 12 L 15 "13.14_ years"

We have now added a space

- P 12 L 17 "and treated public (NHS) hospitals (84.1%), had a mean age of 69.8 years"

We have revised this sentence [Page 12]

- P 16 L 48 no, the National Joint Registry is not the largest in the world!

The National Joint Registry of England Wales, Northern Ireland and the Isle of Man has in excess of 2.5 million hip and knee procedures contained within it. We know of no other country that has a register of this size. We now cite Malchau et al. that indicates that it is the world's largest register. [Page 17]

Reviewer: 3 (Sameer Naranje)

We would like to thank Sameer Naranje for taking the time to review our work.

A very well formulated study. However, authors have not compared the results of their study to the existing similar studies in the discussion section and needs to be revised.

Unfortunately, there are no similar studies which have decomposed the volume association into within- and between- consultants. We now more clearly state that previous analyses are single-level analyses and then describe why they are fundamentally misleading.[Page 14]

"," after before reference citations should be removed.

We will change the positions of the commas in the copy editing process

Reviewer: 4 (Kim Madden)

We would like to thank Kim Madden for taking the time to review our work.

This is a very interesting an important topic in orthopaedics.
The authors have taken great care in conducting this study in a more methodologically stringent way than previous studies and for that I applaud them.

We thank you for recognising the methodological rigor in this article.

It is well-written and reported according to STROBE standards. I do have a few conceptual questions that I am hoping the authors can respond to:

1. For within consultant effects, how do you distinguish whether the volume or the expertise is affecting the outcomes? Although they are related, I think expertise and volume at a particular point in time are two separate concepts.

This is a very interesting and salient point.

Unfortunately, expertise (or experience) is very difficult to characterise. We believe the implicit assumption is the volume of procedures performed mediates expertise. We suggest overall average volume goes some way to estimating this. Therefore the "between-" consultant volume is estimating the expertise of the surgeon.

Unfortunately, totality of procedures is difficult to determine as some surgeons were performing procedures 30 years before the start of the register. Similarly, many surgeons train outside the coverage of the registry, hold fellowships outside of the country before becoming consultants or migrate when already a consultant, and this information is not captured in the registry.

We now discuss the concept of expertise, and the difficulties faced when attempting to characterise it. [page 16]

2. Second line of the discussion: the authors state "...to analyse the effects of consultant volumes on revision...". I think this should be association not effects as the authors themselves acknowledge that they cannot establish a causal relationship.

You are indeed correct, we have now changed this to association. [Page 14]

3. While I agree with the authors that "...these factors are not causally related to volume, but rather due to unmeasured factors such as centre effects.", I don't think it necessarily follows that surgeons should not increase their volume to improve outcomes. What about expertise effects? If surgeons do more cases they will become better at those cases over time. However, if high-volume surgeons decrease their volume (within reason) they will not necessarily lose that level of expertise. I don't think this study has adequately addressed the concept of expertise changing over time.

We believe this expertise/experience is probably best characterised by a surgeon's average surgical volume given the data available to us for analysis and represented in the between-consultant analysis. This is why high average, experienced, volume surgeons have better results than low average inexperienced surgeons. As you suggest it may be somewhat unsurprising that if a high average volume surgeon reduces their volume the surgical results are unaffected. Similarly, if a low volume, inexperienced, surgeon temporarily increases their personal volume it does not alter their results either, i.e. they still remain inexperienced.

This is somewhat akin to state vs. trait behaviour. And whilst it is technically true, that the surgeon's average volume, their "trait", can be modified over an extended period of time, we think that the normally distributed "within surgeon" volume displayed in figure 2, suggests this does not occur frequently. Furthermore, Figure 3 which essentially shows box and whisker plots for each consultant surgeon in the registry illustrates that their IQR are approximately centred on their personal mean, add evidence to this argument.

Furthermore, if expertise were to be maximised, by maximising volume, you would see a positive (reduced revision) within-consultant effect as surgeons increased their volume. The opposite of this appears to be the case, given the non-significant trend is for an increasing revision rate as a surgeon volume increases. It seems surgeons are best working at their average volume.

We believe our data supports your assumption if you consider the between-surgeon results as representing expertise i.e. a more stable trait, and within surgeon volume over the time periods considered is a state.

We now comment on this in the discussion. [Page 17]

4. The authors only considered revision in their analyses and not any other outcomes. This is a limitation, particularly because threshold for revision can change over time as surgeons gain expertise and experience.

We only considered revision, but we agree other outcomes are likely to be of interest. [Page 16]

We also note that it is not always the primary surgeon that conducts the revision. [Page 8]

5. The authors state in the conclusion that "Encouraging consultants to undertake a minimum number of procedures under the guise of raising standards could be counterproductive and may only serve to expose patients to increased risk of revision by low or previously low volume consultants". This might be true if the surgeons have little interest in or aptitude for increasing

their volume. However, I think expertise is affected by both surgeon factors and volume, which isn't really addressed here.

We agree volume and surgeon factors are likely to be important characteristics in developing skills for training surgeons. The consultant surgeons that we are considering here are already trained and working independently. Furthermore, the descriptive data presented in Figures 2 and 3 suggest that personal volume appears to be a fairly stable characteristic.

We believe you are likely to be correct that other surgeon factors may mediate increases in expertise and improve outcomes. However, changes in volume alone are unlikely to precipitate the desired outcomes.

We now comment on this in the discussion [Page 16, 17].

Reviewer: 5 (Zhiying You)

Please leave your comments for the authors below It is a very interesting study with potential implication of policy making. Addressing the following concerns could make it more interesting to readers.

1. Provide more details or more clear definition of the variables involved in the analysis wherever appropriate.

Supplementary Table 1 includes a detailed description of variables and how they are defined, unfortunately there is not space within the word limits for this to be contained in the main paper.

2. Page 10, line 10, while the Weibull distribution was specified for analysis, readers may wonder what if other distributions had been assumed. Furthermore, adding some more details of the survival model used would help readers understand better the methods and make it possible that readers can replicate the analysis for given data.

We found the Weibull distribution has enough flexibility to provide a reasonable approximation to the baseline hazard. We tried to build models using multi-level flexible parametric models, however we encountered issues with numerical overflow. However, we do note that minor mis-specification in the baseline hazard seems to have little effect on the estimates of the effects of primary interest.

We now describe the survival model in more detail in the supplementary material, and point to this in the manuscript [page 11]

3. Page 10, line 55, while it was mentioned that "Given the large data set and small fraction of incomplete cases among the observed (FICO), any improvement of efficiency from multiple imputation is likely to be negligible, and a complete-cases analysis would provide unbiased results", readers may want to know how small the fraction was. Providing the number will be better.

We have now indicated that fraction of complete cases is 89% of the data. [Page 11]

4. Page 14, line 39, it was mentioned "rather due to unmeasured factors such as centre effects". Probably it should also include unmeasured physician and/or patient factors.

We have now amended this to include unmeasured surgeon and/or patient factors. [Page 14]

5. Pages 25-28, there are needs of markers numbers for the figures, e.g. figure 3, figure 4. Right now readers don't know which figure was referred to when reading Figure 2 and Figure 3.

We assume they will be applied in the typesetting process. If this is incorrect, we are happy to add these.

VERSION 2 – REVIEW

REVIEWER	Richard Jenkinson Sunnybrook Health Sciences Center University of Toronto Canada
REVIEW RETURNED	23-Jan-2020

GENERAL COMMENTS	An improved version of a previously reviewed manuscript. The conclusions are more appropriate based on the data presented. The central thesis of the paper is that individual surgeon volume change does not affect revision rates. However, most of the surgeons in the data have volumes higher than previously proposed thresholds for worse complication rates. It makes sense that within surgeon variation from high volume to slightly higher or lower volume would not change revision rates. There is the assertion that low volume surgeons increasing their volume above thresholds would not improve outcomes. Is that assertion testable from the data more directly? If I am missing that, can it be displayed more clearly? Perhaps a sensitivity analysis showing the within surgeon effects for the high versus the low volume surgeons. This would strengthen the conclusions of the paper. Between surgeon hazard ratio crosses one at around 40 in this data. That could be a cut-off point or 35 which has been previously published. Is there a group of surgeons who transitioned from low to higher volume in the data to directly support or refute the assertion that increasing surgical volume did not improve outcomes?
--

REVIEWER	Andreas Halder Sana Kliniken Sommerfeld, Germany
REVIEW RETURNED	20-Feb-2020

GENERAL COMMENTS	 - Thank you for revising the manuscript according to our comments. It is much more focused and clear. - Still statistical method description and graphs are hard to understand for non-statisticians and dominate the orthopaedic content at some points. - There is still no risk adjustment for complexity of procedure or implants used. - Still all cause revision as primary outcome is probably not suitable as soft tissue revision for hematoma is more depending on surgeon's preferences than surgical quality and is very different from revision for dislocation or infection. - The conclusion, that volume has no effect on revision rate is not fully justified, as the time the study looks at volume changes within one surgeon is too short. It should be stated that volume alone does not necessarily have an effect on revision rate alone, as is it might
---

	have an effect on experience over longer period of time. - With this important addition to the conclusion and limitations section I would vote for acceptance.
--	---

REVIEWER	Kim Madden McMaster University, Canada
REVIEW RETURNED	04-Feb-2020

GENERAL COMMENTS	Thank you, I believe that the authors have sufficiently addressed my comments. This is very complex conceptually, but I think the authors have tried to make the paper as user-friendly as possible.
--

REVIEWER	
REVIEW RETURNED	

GENERAL COMMENTS	
--

REVIEWER	Zhiying You Colorado University Anschutz Medical Campus
REVIEW RETURNED	17-Jan-2020

GENERAL COMMENTS	All previous concerns have been solved, except that figure 3, figure 4 and figure 5 cannot be found even they were mentioned in the text.
---

VERSION 2 – AUTHOR RESPONSE

Reviewer: 1

Richard Jenkinson

An improved version of a previously reviewed manuscript. The conclusions are more appropriate based on the data presented.

We thank you for taking the time to read the manuscript and are pleased the conclusions are more appropriate.

The central thesis of the paper is that individual surgeon volume change does not affect revision rates. However, most of the surgeons in the data have volumes higher than previously proposed thresholds for worse complication rates. It makes sense that within surgeon variation from high volume to slightly higher or lower volume would not change revision rates. There is the assertion that low volume surgeons increasing their volume above thresholds would not improve outcomes. Is that assertion testable from the data more directly? If I am missing that, can it be displayed more clearly? Perhaps a sensitivity analysis showing the within surgeon effects for the high versus the low volume surgeons. This would strengthen the conclusions of the paper.

We believe that the within surgeon analysis is the most appropriate and direct way of testing the hypothesis that “changes in an individual surgeons volume effects their outcome”. Whilst it is some what true that some low volume surgeon never become high volume surgeons, there are a number of surgeons with low volumes that do become higher volume surgeons, and conversely there are a number of higher volume surgeons that have low volume periods as a surgeons.

When interpreting the within surgeon effect, we interpret shifts to the right hand side of figure 6 as low volume surgeons increasing their personal volumes, whereas shifts to the left are higher

volume surgeons reducing their volume. This interpretation is borne out of the fact that surgeons with a low volume can not effectively reduce their volume, whereas higher volume surgeons can effectively reduce their volume. We also note that moderate volumes (50-100 procedures) can both increase and decrease their volume.

The reviewer has suggested that a stratified analysis (low versus high volume surgeons) would be helpful in showing this. We suggest the current figures already provide this information. As one can see from both figure 3 most surgeon do not increase or decrease their volume by more than 50 procedures per year. Figure 6 then shows that a shift of +/- 50 procedures for your existing mean is not associated with any benefit or harm as the hazard ratios do not change by more than 5% and the 95% confidence intervals for both the crude and adjusted models cross over the null effect of 1. We therefore see no really added value of showing any further figures. It is important to add that these volume changes are for experienced consultants who have already passed early improvements that one might observe at the trainee level due to practice. Low volume in this context does not reflect lack of experience but rather that these surgeons choose to spend their time doing other procedures thereby maintain their manual dexterity and operative experience. We have added an additional sentence to highlight this point to the discussion.

It is important to add that these volume changes are for experienced consultants who have already passed early improvements that one might observe at the early trainee level due to practice. Furthermore, their low volume for hip replacement may reflect high volume experience for other procedures thereby ensuring manual dexterity albeit for a different operation.

Between surgeon hazard ratio crosses one at around 40 in this data. That could be a cut-off point or 35 which has been previously published.

The hazard ratio crosses unity at 32 procedures annually by virtue of centring the restricted cubic spline. 32 is the mean volume of procedures conducted by surgeons within the NJR. We believed that comparing the between surgeon effect, anchored at the mean was the most defensible decision. We now describe the centring of the RCS at the mean in the methods.

We also do note in the discussion:

“optimal between-consultant results are reached when the consultant volumes in the previous year are approximately 200 procedures.”

This is very much more than the 32 procedures that has previously been suggested.

Is there a group of surgeons who transitioned from low to higher volume in the data to directly support or refute the assertion that increasing surgical volume did not improve outcomes?

As surgeons move through a range of volumes throughout their career, see figure 4, it is very difficult to characterize any of the surgeons as definitively high or definitively low. The same is true to some degree with centres, however there are clearly a group of centres that are consistently high or low volume. Whilst within surgeon volume changes may not alter outcome, these changes may be reflected in between surgeon differences related to other organizational and care factors over and above operative technical skill. Thus low volume surgeons that transition from low to high volumes can and do get better results than the low volume surgeons that never transition to high volume surgeons. This result is reflected as the between surgeon effect, see figure 5.

What allows these surgeons to transition from low volume to high volume is unclear. But, we suggest the data does not support low volume surgeons increasing their personal volume in the hope of improving their outcomes due to greater surgical volume.

Reviewer: 2. Andreas Halder

- Thank you for revising the manuscript according to our comments. It is much more focused and clear.

We thank you for taking the time to read the manuscript, and am glad that it is more focused.

- Still statistical method description and graphs are hard to understand for non-statisticians and dominate the orthopaedic content at some points.

We understand the statistical concepts in this paper are complex. However, we believe without this complexity the unique and novel results would not be accessible.

- There is still no risk adjustment for complexity of procedure or implants used.

Unfortunately procedure complexity is very difficult to assess from a registry perspective. In the earliest iteration of the data collection, there was a collection of complex primary. Complex primary was almost universally ticked, therefore this question had little utility. However, to make the procedures as homogenous as possible we limited the sample set to all patients with the single indication of Osteoarthritis only. Any patients with other reasons indicated was excluded from the analysis sample, but not the volume calculation. This information is reflected in Figure 1.

Accounting for individual implant constructs is very complex, there are 10's of thousands of constructs used within the NJR. In order to make this problem more tractable we do adjust for bearing type, and fixation technique. We believe this is an effective method for controlling for implant choice.

We also acknowledge arguments that suggest implant choice maybe a mediating factor, on the causal pathway, between high and low volume surgeons. For example, Cementless prostheses are typical quicker to fix, therefore you may anticipate higher volume surgeons to have greater use of cementless devices.

However, we appreciate arguments for and against prosthesis adjustment. To these ends we illustrate a wide variety of adjustment from no adjustment, to adjustment for patient characteristics, to adjustment for many different factors including prosthesis. We list all confounding factors in the methods, confound section.

Despite the adjustment used the interpretation of the results is similar.

- Still all cause revision as primary outcome is probably not suitable as soft tissue revision for hematoma is more depending on surgeon's preferences than surgical quality and is very different from revision for dislocation or infection.

We agree other adverse events such as dislocation or infection would be of interest. Unfortunately the data from the joint registry is not the most appropriate way of investigating this as it only records revision of prostheses. As many dislocations maybe reduced in an emergency dept we would be required to link to local hospital data. We hope to explore this useful suggestion more fully in the future.

We do note in the discussion that

“Revision surgery is only one possible endpoint and other endpoints including Patient Reported Outcome Measures (PROMs) or other adverse events may also be important to patients.”

- The conclusion, that volume has no effect on revision rate is not fully justified, as the time the study looks at volume changes within one surgeon is too short. It should be stated that volume alone does not necessarily have an effect on revision rate alone, as is it might have an effect on experience over longer period of time.

In England and Wales, elective arthroplasty can be influenced by winter pressures and seasonality, therefore in order to mitigate this the period of interest must be at least one whole calendar year. If it was not 1 calendar year, the influence of season would impact the volume calculation. However, we would also argue that choosing a period of longer than 1 year would smooth away any signals in rapid changes in volume. I.E. Choosing a 730 day lag from the index procedure would simply smooth away any minor peaks and troughs in volume and result in interpreting data over a biennial period instead of annual period.

However, we do acknowledge that we have not been able to fully capture the experience phenomenon.

We note in the discussion

“ Volume does not fully encapsulate experience or expertise that may have been acquired prior to the inception of the register for example during fellowships or from working in other healthcare systems, or the potential persistent effect of learnt experience.”

- With this important addition to the conclusion and limitations section I would vote for acceptance.

We have now reflected on your points in the discussion, and would like to thank you for taking the time to read and review this manuscript.

Reviewer: 4 Kim Madden

Thank you, I believe that the authors have sufficiently addressed my comments. This is very complex conceptually, but I think the authors have tried to make the paper as user-friendly as possible.

We thank the reviewer to take the time with this paper, we understand it is conceptually difficult.

Reviewer: 5
Zhiying You

All previous concerns have been solved, except that figure 3, figure 4 and figure 5 cannot be found even they were mentioned in the text.

We thank you for taking the time to read the revised version of the manuscript and are pleased we have addressed all of your concerns. Please note the figures for the manuscript have been uploaded separately.

VERSION 3 – REVIEW

REVIEWER	Richard Jenkinson University of Toronto Canada
REVIEW RETURNED	06-Jul-2020
GENERAL COMMENTS	The updated version of the manuscript is a much improved version compared to previous. The complex statistical methods are described more clearly and the figure usage is more streamlined and understandable. The main message of the paper is communicated much more clearly with appropriate caveats. The authors addressed my previous concerns to my satisfaction.